# EXPLAINABLE DEEP ONE-CLASS CLASSIFICATION

**Philipp Liznerski**[1]*     **Lukas Ruff**[2]*     **Robert A. Vandermeulen**[2]*
**Billy Joe Franks**[1]     **Marius Kloft**[1]     **Klaus-Robert Müller**[2][3][4][5]

[1]ML group, Technical University of Kaiserslautern, Germany
[2]ML group, Technical University of Berlin, Germany
[3]Google Research, Brain Team, Berlin, Germany
[4]Department of Artificial Intelligence, Korea University, Seoul, Republic of Korea
[5]Max Planck Institute for Informatics, Saarbrücken, Germany
`{liznerski, franks, kloft}@cs.uni-kl.de`
`{lukas.ruff, vandermeulen, klaus-robert.mueller}@tu-berlin.de`

## ABSTRACT

Deep one-class classification variants for anomaly detection learn a mapping that concentrates nominal samples in feature space causing anomalies to be mapped away. Because this transformation is highly non-linear, finding interpretations poses a significant challenge. In this paper we present an explainable deep one-class classification method, *Fully Convolutional Data Description* (FCDD), where the mapped samples are themselves also an explanation heatmap. FCDD yields competitive detection performance and provides reasonable explanations on common anomaly detection benchmarks with CIFAR-10 and ImageNet. On MVTec-AD, a recent manufacturing dataset offering ground-truth anomaly maps, FCDD sets a new state of the art in the unsupervised setting. Our method can incorporate ground-truth anomaly explanations during training and using even a few of these ($\sim 5$) improves performance significantly. Finally, using FCDD's explanations, we demonstrate the vulnerability of deep one-class classification models to spurious image features such as image watermarks.[1]

## 1 INTRODUCTION

Anomaly detection (AD) is the task of identifying anomalies in a corpus of data (Edgeworth, 1887; Barnett and Lewis, 1994; Chandola et al., 2009; Ruff et al., 2021). Powerful new anomaly detectors based on deep learning have made AD more effective and scalable to large, complex datasets such as high-resolution images (Ruff et al., 2018; Bergmann et al., 2019). While there exists much recent work on deep AD, there is limited work on making such techniques explainable. Explanations are needed in industrial applications to meet safety and security requirements (Berkenkamp et al., 2017; Katz et al., 2017; Samek et al., 2020), avoid unfair social biases (Gupta et al., 2018), and support human experts in decision making (Jarrahi, 2018; Montavon et al., 2018; Samek et al., 2020). One typically makes anomaly detection explainable by annotating pixels with an anomaly score and, in some applications, such as finding tumors in cancer detection (Quellec et al., 2016), these annotations are the primary goal of the detector.

One approach to deep AD, known as *Deep Support Vector Data Description* (DSVDD) (Ruff et al., 2018), is based on finding a neural network that transforms data such that nominal data is concentrated to a predetermined center and anomalous data lies elsewhere. In this paper we present *Fully Convolutional Data Description* (FCDD), a modification of DSVDD so that the transformed samples are themselves an image corresponding to a downsampled anomaly heatmap. The pixels in this heatmap that are far from the center correspond to anomalous regions in the input image. FCDD does this by only using convolutional and pooling layers, thereby limiting the receptive field of each output pixel. Our method is based on the one-class classification paradigm (Moya et al., 1993; Tax, 2001; Tax and Duin, 2004; Ruff et al., 2018), which is able to naturally incorporate known anomalies Ruff et al. (2021), but is also effective when simply using synthetic anomalies.

---

*equal contribution

[1]Our code is available at: `https://github.com/liznerski/fcdd`

We show that FCDD's anomaly detection performance is close to the state of the art on the standard AD benchmarks with CIFAR-10 and ImageNet while providing transparent explanations. On MVTec-AD, an AD dataset containing ground-truth anomaly maps, we demonstrate the accuracy of FCDD's explanations (see Figure 1), where FCDD sets a new state of the art. In further experiments we find that deep one-class classification models (e.g. DSVDD) are prone to the "Clever Hans" effect (Lapuschkin et al., 2019) where a detector fixates on spurious features such as image watermarks. In general, we find that the generated anomaly heatmaps are less noisy and provide more structure than the baselines, including gradient-based methods (Simonyan et al., 2013; Sundararajan et al., 2017) and autoencoders (Sakurada and Yairi, 2014; Bergmann et al., 2019).

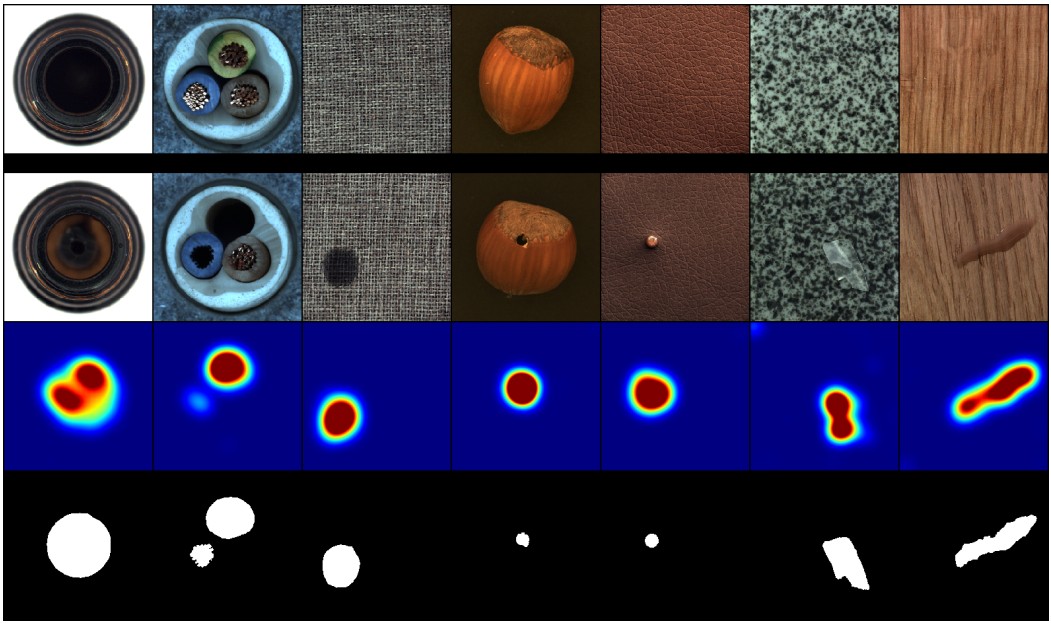

Figure 1: FCDD explanation heatmaps for MVTec-AD (Bergmann et al., 2019). Rows from top to bottom show: (1) nominal samples (2) anomalous samples (3) FCDD anomaly heatmaps (4) ground-truth anomaly maps.

## 2 RELATED WORK

Here we outline related works on deep AD focusing on explanation approaches. Classically deep AD used autoencoders (Hawkins et al., 2002; Sakurada and Yairi, 2014; Zhou and Paffenroth, 2017; Zhao et al., 2017). Trained on a nominal dataset autoencoders are assumed to reconstruct anomalous samples poorly. Thus, the reconstruction error can be used as an anomaly score and the pixel-wise difference as an explanation (Bergmann et al., 2019), thereby naturally providing an anomaly heatmap. Recent works have incorporated attention into reconstruction models that can be used as explanations (Venkataramanan et al., 2019; Liu et al., 2020). In the domain of videos, Sabokrou et al. (2018) used a pre-trained fully convolutional architecture in combination with a sparse autoencoder to extract 2D features and provide bounding boxes for anomaly localization. One drawback of reconstruction methods is that they offer no natural way to incorporate known anomalies during training.

More recently, one-class classification methods for deep AD have been proposed. These methods attempt to separate nominal samples from anomalies in an unsupervised manner by concentrating nominal data in feature space while mapping anomalies to distant locations (Ruff et al., 2018; Chalapathy et al., 2018; Goyal et al., 2020). In the domain of NLP, DSVDD has been successfully applied to text, which yields a form of interpretation using attention mechanisms (Ruff et al., 2019). For images, Kauffmann et al. (2020) have used a deep Taylor decomposition (Montavon et al., 2017) to derive relevance scores.

Some of the best performing deep AD methods are based on self-supervision. These methods transform nominal samples, train a network to predict which transformation was used on the input, and

provide an anomaly score via the confidence of the prediction (Golan and El-Yaniv, 2018; Hendrycks et al., 2019b). Hendrycks et al. (2019a) have extended this to incorporate known anomalies as well. No explanation approaches have been considered for these methods so far.

Finally, there exists a great variety of explanation methods in general, for example model-agnostic methods (e.g. LIME (Ribeiro et al., 2016)) or gradient-based techniques (Simonyan et al., 2013; Sundararajan et al., 2017). Relating to our work, we note that fully convolutional architectures have been used for *supervised* segmentation tasks where target segmentation maps are required during training (Long et al., 2015; Noh et al., 2015).

## 3 EXPLAINING DEEP ONE-CLASS CLASSIFICATION

We review one-class classification and fully convolutional architectures before presenting our method.

**Deep One-Class Classification** Deep one-class classification (Ruff et al., 2018; 2020b) performs anomaly detection by learning a neural network to map nominal samples near a center $\mathbf{c}$ in output space, causing anomalies to be mapped away. For our method we use a *Hypersphere Classifier* (HSC) (Ruff et al., 2020a), a recently proposed modification of Deep SAD (Ruff et al., 2020b), a semi-supervised version of DSVDD (Ruff et al., 2018). Let $X_1, \ldots, X_n$ denote a collection of samples and $y_1, \ldots, y_n$ be labels where $y_i = 1$ denotes an anomaly and $y_i = 0$ denotes a nominal sample. Then the HSC objective is

$$\min_{\mathcal{W},\mathbf{c}} \quad \frac{1}{n}\sum_{i=1}^{n}(1-y_i)h(\phi(X_i;\mathcal{W})-\mathbf{c}) - y_i \log\left(1 - \exp\left(-h(\phi(X_i;\mathcal{W})-\mathbf{c})\right)\right), \qquad (1)$$

where $\mathbf{c} \in \mathbb{R}^d$ is the center, and $\phi : \mathbb{R}^{c\times h\times w} \to \mathbb{R}^d$ a neural network with weights $\mathcal{W}$. Here $h$ is the pseudo-Huber loss (Huber et al., 1964), $h(\mathbf{a}) = \sqrt{\|\mathbf{a}\|_2^2 + 1} - 1$, which is a robust loss that interpolates from quadratic to linear penalization. The HSC loss encourages $\phi$ to map nominal samples near $\mathbf{c}$ and anomalous samples away from the center $\mathbf{c}$. In our implementation, the center $\mathbf{c}$ corresponds to the bias term in the last layer of our networks, i.e. is included in the network $\phi$, which is why we omit $\mathbf{c}$ in the FCDD objective below.

**Fully Convolutional Architecture** Our method uses a *fully convolutional network* (FCN) (Long et al., 2015; Noh et al., 2015) that maps an image to a matrix of features, i.e. $\phi : \mathbb{R}^{c\times h\times w} \to \mathbb{R}^{1\times u\times v}$ by using alternating convolutional and pooling layers only, and does not contain any fully connected layers. In this context, pooling can be seen as a special kind of convolution with fixed parameters.

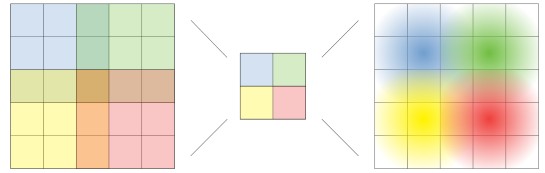

Figure 2: Visualization of a 3×3 convolution followed by a 3×3 transposed convolution with a Gaussian kernel, both using a stride of 2.

A core property of a convolutional layer is that each pixel of its output only depends on a small region of its input, known as the output pixel's *receptive field*. Since the output of a convolution is produced by moving a filter over the input image, each output pixel has the same relative position as its associated receptive field in the input. For instance, the lower-left corner of the output representation has a corresponding receptive field in the lower-left corner of the input image, etc. (see Figure 2 left side). The outcome of several stacked convolutions also has receptive fields of limited size and consistent relative position, though their size grows with the amount of layers. Because of this an FCN preserves spatial information.

**Fully Convolutional Data Description** Here we introduce our novel explainable AD method *Fully Convolutional Data Description* (FCDD). By taking advantage of FCNs along with the HSC above, we propose a deep one-class method where the output features preserve spatial information and also serve as a downsampled anomaly heatmap. For situations where one would like to have a full-resolution heatmap, we include a methodology for upsampling the low-resolution heatmap based on properties of receptive fields.

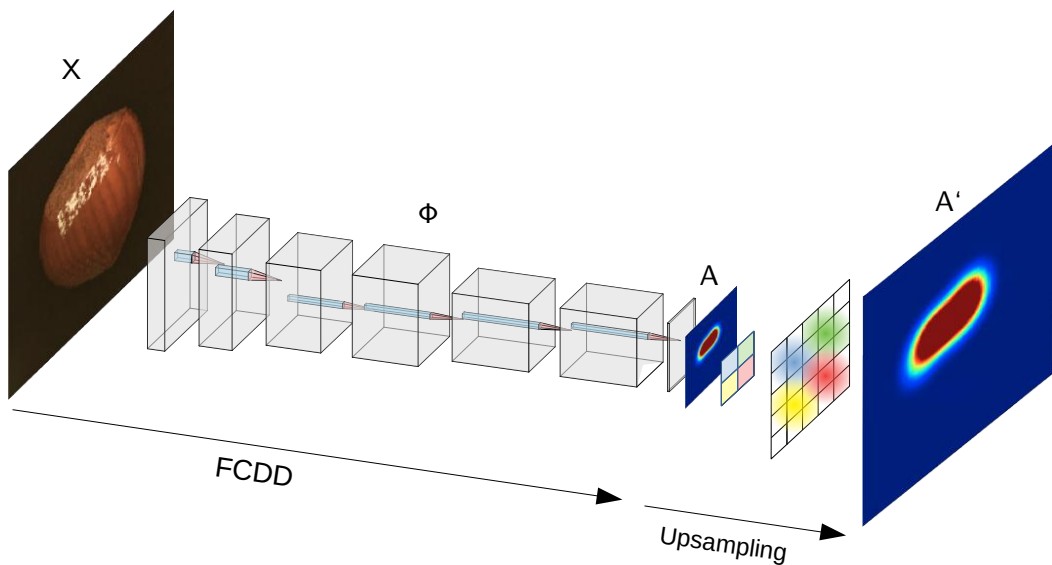

Figure 3: Visualization of the overall procedure to produce full-resolution anomaly heatmaps with FCDD. $X$ denotes the input, $\phi$ the network, $A$ the produced anomaly heatmap and $A'$ the upsampled version of $A$ using a transposed Gaussian convolution.

FCDD is trained using samples that are labeled as nominal or anomalous. As before, let $X_1, \ldots, X_n$ denote a collection of samples with labels $y_1, \ldots, y_n$ where $y_i = 1$ denotes an anomaly and $y_i = 0$ denotes a nominal sample. Anomalous samples can simply be a collection of random images which are not from the nominal collection, e.g. one of the many large collections of images which are freely available like 80 Million Tiny Images (Torralba et al., 2008) or ImageNet (Deng et al., 2009). The use of such an auxiliary corpus has been recommended in recent works on deep AD, where it is termed *Outlier Exposure* (OE) (Hendrycks et al., 2019a;b). When one has access to "true" examples of the anomalous dataset, i.e. something that is likely to be representative of what will be seen at test time, we find that even using a few examples as the corpus of labeled anomalies performs exceptionally well. Furthermore, in the absence of *any* sort of known anomalies, one can generate synthetic anomalies, which we find is also very effective.

With an FCN $\phi : \mathbb{R}^{c \times h \times w} \to \mathbb{R}^{u \times v}$ the FCDD objective utilizes a pseudo-Huber loss on the FCN output matrix $A(X) = \left( \sqrt{\phi(X; \mathcal{W})^2 + 1} - 1 \right)$, where all operations are applied element-wise. The FCDD objective is then defined as (cf., (1)):

$$\min_{\mathcal{W}} \quad \frac{1}{n} \sum_{i=1}^{n} (1 - y_i) \frac{1}{u \cdot v} \|A(X_i)\|_1 - y_i \log \left( 1 - \exp \left( -\frac{1}{u \cdot v} \|A(X_i)\|_1 \right) \right). \tag{2}$$

Here $\|A(X)\|_1$ is the sum of all entries in $A(X)$, which are all positive. FCDD is the utilization of an FCN in conjunction with the novel adaptation of the HSC loss we propose in (2). The objective maximizes $\|A(X)\|_1$ for anomalies and minimizes it for nominal samples, thus we use $\|A(X)\|_1$ as the anomaly score. Entries of $A(X)$ that contribute to $\|A(X)\|_1$ correspond to regions of the input image that add to the anomaly score. The shape of these regions depends on the receptive field of the FCN. We include a sensitivity analysis on the size of the receptive field in Appendix A, where we find that performance is not strongly affected by the receptive field size. Note that $A(X)$ has spatial dimensions $u \times v$ and is smaller than the original image dimensions $h \times w$. One could use $A(X)$ directly as a low-resolution heatmap of the image, however it is often desirable to have full-resolution heatmaps. Because we generally lack ground-truth anomaly maps in an AD setting during training, it is not possible to train an FCN in a supervised way to upsample the low-resolution heatmap $A(X)$ (e.g. as in (Noh et al., 2015)). For this reason we introduce an upsampling scheme based on the properties of receptive fields.

**Heatmap Upsampling** Since we generally do not have access to ground-truth pixel annotations in anomaly detection during training, we cannot learn how to upsample using a deconvolutional type of structure. We derive a principled way to upsample our lower resolution anomaly heatmap instead. For every output pixel in $A(X)$ there is a unique input pixel which lies at the center of its receptive field. It has been observed before that the effect of the receptive field for an output pixel decays in a Gaussian manner as one moves away from the center of the receptive field (Luo et al.,

---

**Algorithm 1** Receptive Field Upsampling

**Input:** $A \in \mathbb{R}^{u \times v}$ (low-res anomaly heatmap)

**Output:** $A' \in \mathbb{R}^{h \times w}$ (full-res anomaly heatmap)

**Define:** $[G_2(\mu, \sigma)]_{x,y} \triangleq \frac{1}{2\pi\sigma^2} \exp\left(-\frac{(x-\mu_1)^2 + (y-\mu_2)^2}{2\sigma^2}\right)$

$A' \leftarrow 0$
**for all** output pixels $a$ in $A$ **do**
$\quad f \leftarrow$ receptive field of $a$
$\quad c \leftarrow$ center of field $f$
$\quad A' \leftarrow A' + a \cdot G_2(c, \sigma)$
**end for**
**return** $A'$

---

2016). We use this fact to upsample $A(X)$ by using a strided transposed convolution with a fixed Gaussian kernel (see Figure 2 right side). We describe this operation and procedure in Algorithm 1 which simply corresponds to a strided transposed convolution. The kernel size is set to the receptive field range of FCDD and the stride to the cumulative stride of FCDD. The variance of the distribution can be picked empirically (see Appendix B for details). Figure 3 shows a complete overview of our FCDD method and the process of generating full-resolution anomaly heatmaps.

## 4 EXPERIMENTS

In this section, we experimentally evaluate the performance of FCDD both quantitatively and qualitatively. For a quantitative evaluation, we use the Area Under the ROC Curve (AUC) (Spackman, 1989) which is the commonly used measure in AD. For a qualitative evaluation, we compare the heatmaps produced by FCDD to existing deep AD explanation methods. As baselines, we consider gradient-based methods (Simonyan et al., 2013) applied to hypersphere classifier (HSC) models (Ruff et al., 2020a) with unrestricted network architectures (i.e. networks that also have fully connected layers) and autoencoders (Bergmann et al., 2019) where we directly use the pixel-wise reconstruction error as an explanation heatmap. We slightly blur the heatmaps of the baselines with the same Gaussian kernel we use for FCDD, which we found results in less noisy, more interpretable heatmaps. We include heatmaps without blurring in Appendix G. We adjust the contrast of the heatmaps per method to highlight interesting features; see Appendix C for details. For our experiments we don't consider model-agnostic explanations, such as LIME (Ribeiro et al., 2016) or anchors (Ribeiro et al., 2018), because they are not tailored to the AD task and performed poorly.

### 4.1 STANDARD ANOMALY DETECTION BENCHMARKS

We first evaluate FCDD on the Fashion-MNIST, CIFAR-10, and ImageNet datasets. The common AD benchmark is to utilize these classification datasets in a one-vs-rest setup where the "one" class is used as the nominal class and the rest of the classes are used as anomalies at test time. For training, we only use nominal samples as well as random samples from some auxiliary Outlier Exposure (OE) (Hendrycks et al., 2019a) dataset, which is separate from the ground-truth anomaly classes following Hendrycks et al. (2019a;b). We report the mean AUC over all classes for each dataset.

**Fashion-MNIST** We consider each of the ten Fashion-MNIST (Xiao et al., 2017) classes in a one-vs-rest setup. We train Fashion-MNIST using EMNIST (Cohen et al., 2017) or grayscaled CIFAR-100 (Krizhevsky et al., 2009) as OE. We found that the latter slightly outperforms the former ($\sim$3 AUC percent points). On Fashion-MNIST, we use a network that consists of three convolutional layers with batch normalization, separated by two downsampling pooling layers.

**CIFAR-10** We consider each of the ten CIFAR-10 (Krizhevsky et al., 2009) classes in a one-vs-rest setup. As OE we use CIFAR-100, which does not share any classes with CIFAR-10. We use a model similar to LeNet-5 (LeCun et al., 1998), but decrease the kernel size to three, add batch normalization, and replace the fully connected layers and last max-pool layer with two further convolutions.

**ImageNet**  We consider 30 classes from ImageNet1k (Deng et al., 2009) for the one-vs-rest setup following Hendrycks et al. (2019a). For OE we use ImageNet22k with ImageNet1k classes removed (Hendrycks et al., 2019a). We use an adaptation of VGG11 (Simonyan and Zisserman, 2015) with batch normalization, suitable for inputs resized to 224×224 (see Appendix D for model details).

**State-of-the-art Methods**  We report results from state-of-the-art deep anomaly detection methods. Methods that do not incorporate known anomalies are the autoencoder (AE), DSVDD (Ruff et al., 2018), Geometric Transformation based AD (GEO) (Golan and El-Yaniv, 2018), and a variant of GEO by Hendrycks et al. (2019b) (GEO+). Methods that use OE are a Focal loss classifier (Hendrycks et al., 2019b), also GEO+, Deep SAD (Ruff et al., 2020b), and HSC (Ruff et al., 2020a).

Table 1: Mean AUC (over all classes and 5 seeds per class) for Fashion-MNIST, CIFAR-10, and ImageNet. Results from existing literature are marked with an asterisk (Bergman and Hoshen, 2020; Golan and El-Yaniv, 2018; Hendrycks et al., 2019b; Ruff et al., 2020a).

| Dataset | without OE | | | | with OE | | | | |
| | AE | DSVDD* | GEO* | Geo+* | Focal* | Geo+* | Deep SAD* | HSC* | FCDD |
|---|---|---|---|---|---|---|---|---|---|
| Fashion-MNIST | 0.82 | 0.93 | 0.94 | × | × | × | × | × | 0.89 |
| CIFAR-10 | 0.59* | 0.65 | 0.86 | 0.90 | 0.87 | 0.96 | 0.95 | 0.96 | 0.95 |
| ImageNet | 0.56 | × | × | × | 0.56 | 0.86 | 0.97 | 0.97 | 0.94 |

**Quantitative Results**  The mean AUC detection performance on the three AD benchmarks are reported in Table 1. We can see that FCDD, despite using a restricted FCN architecture to improve explainability, achieves a performance that is close to state-of-the-art methods and outperforms autoencoders, which yield a detection performance close to random on more complex datasets. We provide detailed results for all individual classes in Appendix F.

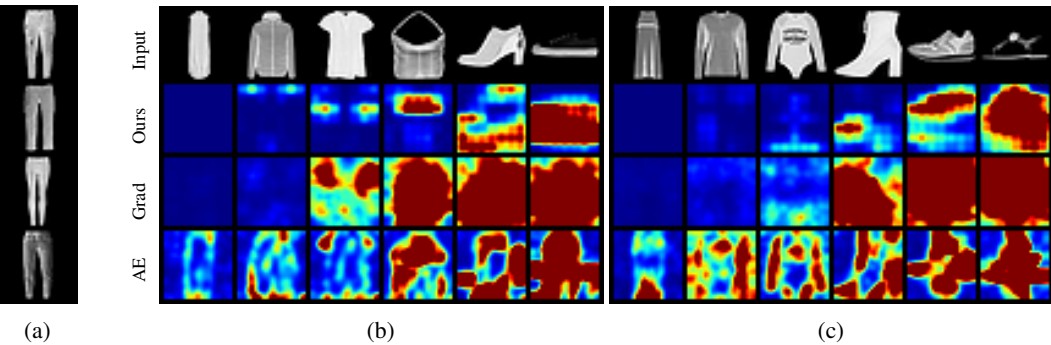

(a)                          (b)                          (c)

Figure 4: Anomaly heatmaps for *anomalous* test samples of a Fashion-MNIST model trained on nominal class "trousers" (nominal samples are shown in (a)). In (b) CIFAR-100 was used for OE and in (c) EMNIST. Columns are ordered by increasing anomaly score from left to right, i.e. what FCDD finds the most nominal looking anomaly on the left to the most anomalous looking anomaly on the right.

**Qualitative Results**  Figures 4 and 5 show the heatmaps for Fashion-MNIST and ImageNet respectively. For a Fashion-MNIST model trained on the nominal class "trousers," the heatmaps show that FCDD correctly highlights horizontal elements as being anomalous, which makes sense since trousers are vertically aligned. For an ImageNet model trained on the nominal class "acorns," we observe that colors seem to be fairly relevant features with green and brown areas tending to be seen as more nominal, and other colors being deemed anomalous, for example the red barn or the white snow. Nonetheless, the method also seems capable of using more semantic features, for example it recognizes the green caterpillar as being anomalous and it distinguishes the acorn to be nominal despite being against a red background.

Figure 6 shows heatmaps for CIFAR-10 models with varying amount of OE, all trained on the nominal class "airplane." We can see that, as the number of OE samples increases, FCDD tends to concentrate

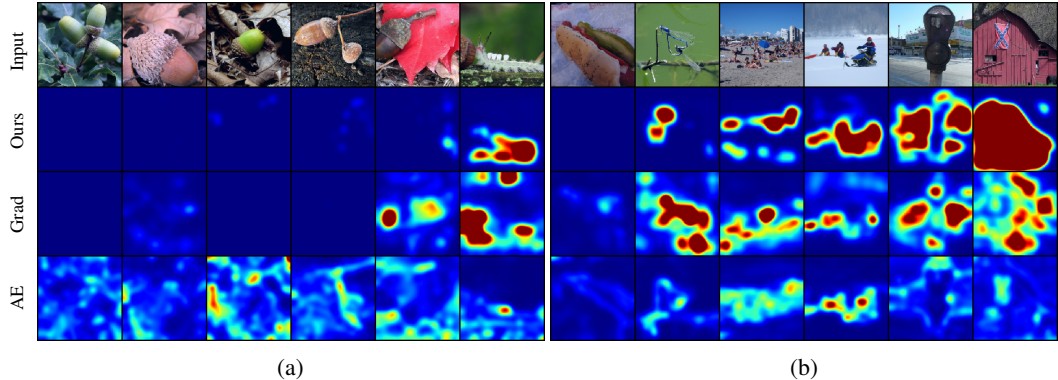

Figure 5: Anomaly heatmaps of an ImageNet model trained on nominal class "acorns." Here (a) are nominal samples and (b) are anomalous samples. Columns are ordered by increasing anomaly score from left to right, i.e. what FCDD finds the most nominal looking on the left to the most anomalous looking on the right for (a) nominal samples and (b) anomalies.

the explanations more on the primary object in the image, i.e. the bird, ship, and truck. We provide further heatmaps for additional classes from all datasets in Appendix G.

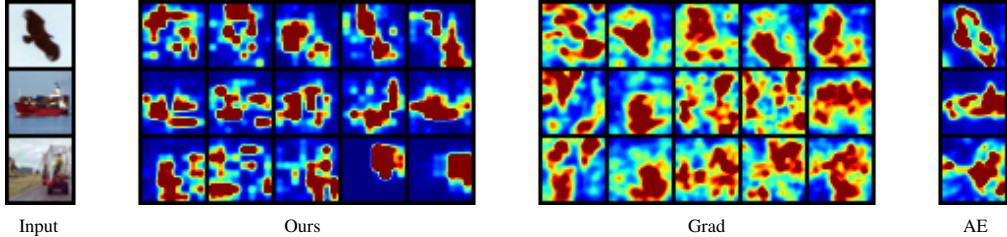

Figure 6: Anomaly heatmaps for three anomalous test samples on a CIFAR-10 model trained on nominal class "airplane." The second, third, and fourth blocks show the heatmaps of FCDD, gradient-based heatmaps of HSC, and AE heatmaps respectively. For Ours and Grad, we grow the number of OE samples from 2, 8, 128, 2048 to full OE. AE is not able to incorporate OE.

**Baseline Explanations**   We found the gradient-based heatmaps to mostly produce centered blobs which lack spatial context (see Figure 6) and thus are not useful for explaining. The AE heatmaps, being directly tied to the reconstruction error anomaly score, look reasonable. We again note, however, that it is not straightforward how to include auxiliary OE samples or labeled anomalies into an AE approach, which leaves them with a poorer detection performance (see Table 1). Overall we find that the proposed FCDD anomaly heatmaps yield a good and consistent visual interpretation.

## 4.2   EXPLAINING DEFECTS IN MANUFACTURING

Here we compare the performance of FCDD on the MVTec-AD dataset of defects in manufacturing (Bergmann et al., 2019). This datasets offers annotated ground-truth anomaly segmentation maps for testing, thus allowing a quantitative evaluation of model explanations. MVTec-AD contains 15 object classes of high-resolution RGB images with up to $1024 \times 1024$ pixels, where anomalous test samples are further categorized in up to 8 defect types, depending on the class. We follow Bergmann et al. (2019) and compute an AUC from the heatmap pixel scores, using the given (binary) anomaly segmentation maps as ground-truth pixel labels. We then report the mean over all samples of this "explanation" AUC for a quantitative evaluation. For FCDD, we use a network that is based on a VGG11 network pre-trained on ImageNet, where we freeze the first ten layers, followed by additional fully convolutional layers that we train.

**Synthetic Anomalies**  OE with a natural image dataset like ImageNet is not informative for MVTec-AD since anomalies here are subtle defects of the nominal class, rather than being out of class (see Figure 1). For this reason, we generate synthetic anomalies using a sort of "confetti noise," a simple noise model that inserts colored blobs into images and reflects the local nature of anomalies. See Figure 7 for an example.

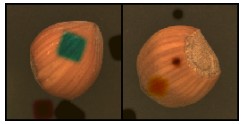

Figure 7: Confetti noise.

**Semi-Supervised FCDD**  A major advantage of FCDD in comparison to reconstruction-based methods is that it can be readily used in a semi-supervised AD setting (Ruff et al., 2020b). To see the effect of having even only a few labeled anomalies and their corresponding ground-truth anomaly maps available for training, we pick for each MVTec-AD class just *one* true anomalous sample per defect type at random and add it to the training set. This results in only 3–8 anomalous training samples. To also take advantage of the ground-truth heatmaps, we train a model on a pixel level. Let $X_1, \ldots, X_n$ again denote a batch of inputs with corresponding ground-truth heatmaps $Y_1, \ldots, Y_n$, each having $m = h \cdot w$ number of pixels. Let $A(X)$ also again denote the corresponding output anomaly heatmap of $X$. Then, we can formulate a pixel-wise objective by the following:

$$\min_{\mathcal{W}} \frac{1}{n} \sum_{i=1}^{n} \left( \frac{1}{m} \sum_{j=1}^{m} (1 - (Y_i)_j) A'(X_i)_j \right) - \log \left( 1 - \exp \left( -\frac{1}{m} \sum_{j=1}^{m} (Y_i)_j A'(X_i)_j \right) \right). \quad (3)$$

**Results**  Figure 1 in the introduction shows heatmaps of FCDD trained on MVTec-AD. The results of the quantitative explanation are shown in Table 2. We can see that FCDD outperforms its competitors in the unsupervised setting and sets a new state of the art of 0.92 pixel-wise mean AUC. In the semi-supervised setting —using only one anomalous sample with corresponding anomaly map per defect class— the explanation performance improves further to 0.96 pixel-wise mean AUC. FCDD also has the most consistent performance across classes.

Table 2: Pixel-wise mean AUC scores for all classes of the MVTec-AD dataset (Bergmann et al., 2019). For competitors we include the baselines presented in the original MVTec-AD paper and previously published works from peer-reviewed venues that include the MVTec-AD benchmark. The competitors are Self-Similarity and L2 Autoencoder (Bergmann et al., 2019), AnoGAN (Schlegl et al., 2017; Bergmann et al., 2019), CNN Feature Dictionaries (Napoletano et al., 2018; Bergmann et al., 2019), Visually Explained Variational Autoencoder (Liu et al., 2020), Superpixel Masking and Inpainting (Li et al., 2020), Gradient Descent Reconstruction with VAEs (Dehaene et al., 2020), and Encoding Structure-Texture Relation with P-Net for AD (Zhou et al., 2020).

| | | | | | unsupervised | | | | | semi-supervised |
|---|---|---|---|---|---|---|---|---|---|---|
| | AE-SS* | AE-L2* | AnoGAN* | CNNFD* | VEVAE* | SMAI* | GDR* | P-NET* | FCDD | FCDD |
| Bottle | 0.93 | 0.86 | 0.86 | 0.78 | 0.87 | 0.86 | 0.92 | **0.99** | 0.97 | 0.96 |
| Cable | 0.82 | 0.86 | 0.78 | 0.79 | 0.90 | **0.92** | 0.91 | 0.70 | 0.90 | **0.93** |
| Capsule | **0.94** | 0.88 | 0.84 | 0.84 | 0.74 | 0.93 | 0.92 | 0.84 | 0.93 | **0.95** |
| Carpet | 0.87 | 0.59 | 0.54 | 0.72 | 0.78 | 0.88 | 0.74 | 0.57 | **0.96** | **0.99** |
| Grid | 0.94 | 0.90 | 0.58 | 0.59 | 0.73 | 0.97 | 0.96 | **0.98** | 0.91 | 0.95 |
| Hazelnut | 0.97 | 0.95 | 0.87 | 0.72 | **0.98** | 0.97 | **0.98** | 0.97 | 0.95 | 0.97 |
| Leather | 0.78 | 0.75 | 0.64 | 0.87 | 0.95 | 0.86 | 0.93 | 0.89 | **0.98** | **0.99** |
| Metal Nut | 0.89 | 0.86 | 0.76 | 0.82 | **0.94** | 0.92 | 0.91 | 0.79 | **0.94** | **0.98** |
| Pill | 0.91 | 0.85 | 0.87 | 0.68 | 0.83 | 0.92 | **0.93** | 0.91 | 0.81 | **0.97** |
| Screw | 0.96 | 0.96 | 0.80 | 0.87 | 0.97 | 0.96 | 0.95 | **1.00** | 0.86 | 0.93 |
| Tile | 0.59 | 0.51 | 0.50 | 0.93 | 0.80 | 0.62 | 0.65 | **0.97** | 0.91 | **0.98** |
| Toothbrush | 0.92 | 0.93 | 0.90 | 0.77 | 0.94 | 0.96 | **0.99** | **0.99** | 0.94 | 0.95 |
| Transistor | 0.90 | 0.86 | 0.80 | 0.66 | **0.93** | 0.85 | 0.92 | 0.82 | 0.88 | 0.90 |
| Wood | 0.73 | 0.73 | 0.62 | 0.91 | 0.77 | 0.80 | 0.84 | **0.98** | 0.88 | 0.94 |
| Zipper | 0.88 | 0.77 | 0.78 | 0.76 | 0.78 | 0.90 | 0.87 | 0.90 | **0.92** | **0.98** |
| Mean | 0.86 | 0.82 | 0.74 | 0.78 | 0.86 | 0.89 | 0.89 | 0.89 | **0.92** | **0.96** |
| $\sigma$ | 0.10 | 0.13 | 0.13 | 0.10 | 0.09 | 0.09 | 0.09 | 0.12 | **0.04** | **0.02** |

## 4.3 THE CLEVER HANS EFFECT

Lapuschkin et al. (2016; 2019) revealed that roughly one fifth of all horse images in PASCAL VOC (Everingham et al., 2010) contain a watermark in the lower left corner. They showed that a classifier recognizes this as the relevant class pattern and fails if the watermark is removed. They call this the

"Clever Hans" effect in memory of the horse Hans, who could correctly answer math problems by reading its master[2]. We adapt this experiment to one-class classification by swapping our standard setup and train FCDD so that the "horse" class is anomalous and use ImageNet as nominal samples. We choose this setup so that one would expect FCDD to highlight horses in its heatmaps and so that any other highlighting makes FCDD reveal a Clever Hans effect.

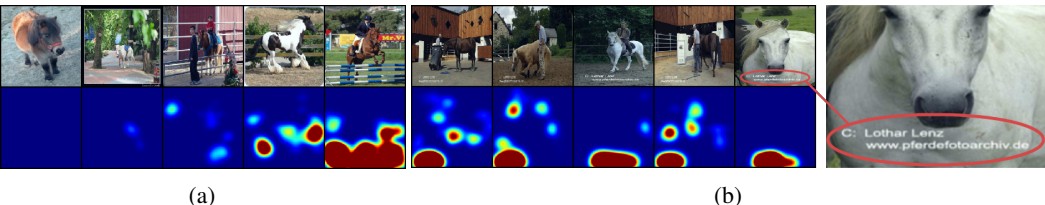

(a)  (b)

Figure 8: Heatmaps for horses on PASCAL VOC. Here (a) shows anomalous samples ordered from most nominal to most anomalous from left to right, and (b) shows examples that indicate that the model is a "Clever Hans," i.e. has learned a characterization based on spurious features (watermarks).

Figure 8 (b) shows that a one-class model is indeed also vulnerable to learning a characterization based on spurious features: the watermarks in the lower left corner which have high scores whereas other regions have low scores. We also observe that the model yields high scores for bars, grids, and fences in Figure 8 (a). This is due to many images in the dataset containing horses jumping over bars or being in fenced areas. In both cases, the horse features themselves do not attain the highest scores because the model has no way of knowing that the spurious features, while providing good discriminative power at training time, would not be desirable upon deployment/test time. In contrast to traditional black-box models, however, transparent detectors like FCDD enable a practitioner to recognize and remedy (e.g. by cleaning or extending the training data) such behavior or other undesirable phenomena (e.g. to avoid unfair social bias).

## 5  CONCLUSION

In conclusion we find that FCDD, in comparison to previous methods, performs well and is adaptable to *both* semantic detection tasks (Section 4.1) and more subtle defect detection tasks (Section 4.2). Finally, directly tying an explanation to the anomaly score should make FCDD less vulnerable to attacks (Anders et al., 2020) in contrast to a posteriori explanation methods. We leave an analysis of this phenomenon for future work.

## ACKNOWLEDGEMENTS

MK, PL, and BJF acknowledge support by the German Research Foundation (DFG) award KL 2698/2-1 and by the German Federal Ministry of Science and Education (BMBF) awards 01IS18051A, 031B0770E, and 01MK20014U. LR acknowledges support by the German Federal Ministry of Education and Research (BMBF) in the project ALICE III (01IS18049B). RV acknowledges support by the Berlin Institute for the Foundations of Learning and Data (BIFOLD) sponsored by the German Federal Ministry of Education and Research (BMBF). KRM was supported in part by the Institute of Information & Communications Technology Planning & Evaluation (IITP) grants funded by the Korea Government (No. 2017-0-00451 and 2019-0-00079) and was partly supported by the German Federal Ministry of Education and Research (BMBF) for the Berlin Center for Machine Learning (01IS18037A-I) and under the Grants 01IS14013A-E, 01GQ1115, 01GQ0850, 01IS18025A, and 031L0207A-D; the German Research Foundation (DFG) under Grant Math+, EXC 2046/1, Project ID 390685689. Finally, we thank all reviewers for their constructive feedback, which helped to improve this work.

---

[2]https://en.wikipedia.org/wiki/Clever_Hans

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

## A RECEPTIVE FIELD SENSITIVITY ANALYSIS

The receptive field has an impact on both detection performance and explanation quality. Here we provide some heatmaps and AUC scores for networks with different receptive field sizes. We observe that the detection performance is only minimally affected, but larger receptive fields cause the explanation heatmap to become less concentrated and more "blobby." For MVTec-AD we see that this can also negatively affect pixel-wise AUC scores, see Table 4.

**CIFAR-10** For CIFAR-10 we create eight different network architectures to study the impact of the receptive field size. Each architecture has four convolutional layers and two max-pool layers. To change the receptive field we vary the kernel size of the first convolutional layer between 3 and 17. When this kernel size is 3 then the receptive field contains approximately one quarter of the image; for a kernel size of 17 the receptive field is the entire image. Table 3 shows the detection performance of the networks. Figure 9 contains example heatmaps.

Table 3: Mean AUC (over all classes and 5 seeds per class) for CIFAR-10 and neural networks with varying receptive field size.

| Receptive field size | 18 | 20 | 22 | 24 | 26 | 28 | 30 | 32 |
|---|---|---|---|---|---|---|---|---|
| AUC | 0.9328 | 0.9349 | 0.9344 | 0.9320 | 0.9303 | 0.9283 | 0.9257 | 0.9235 |

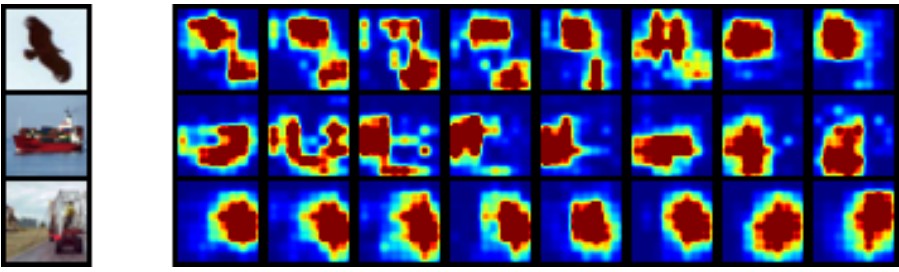

Figure 9: Anomaly heatmaps for three anomalous test samples on CIFAR-10 models trained on nominal class "airplane." We grow the receptive field size from 18 (left) to 32 (right).

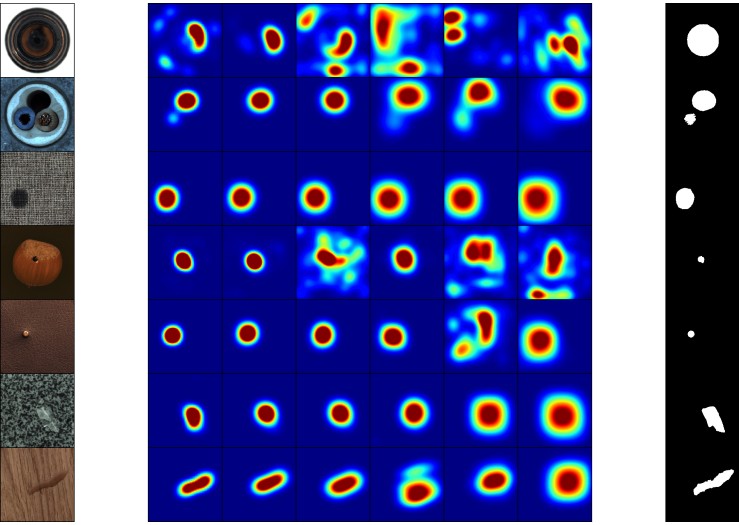

Figure 10: Anomaly heatmaps for seven anomalous test samples of MVTec-AD. We grow the receptive field size from 53 (left) to 243 (right).

Table 4: Pixel-wise mean AUC (over all classes and 5 seeds per class) for MVTec-AD and neural networks with varying receptive field size.

| Receptive field size | 53 | 91 | 129 | 167 | 205 | 243 |
|---|---|---|---|---|---|---|
| AUC | 0.88 | 0.85 | 0.79 | 0.76 | 0.75 | 0.75 |

**MVTec-AD** We create six different network architectures for MVTec-AD. They have six convolutional layers and three max-pool layers. We vary the kernel size for all of the convolutional layers between 3 and 13, which corresponds to a receptive field containing $1/16$ of the image to the full image respectively. Table 4 shows the explanation performance of the networks in terms of pixel-wise mean AUC. Figure 10 contains some example heatmaps. We observe that a smaller receptive field yields better explanation performance.

## B    IMPACT OF THE GAUSSIAN VARIANCE

Using the proposed heatmap upsampling in Section 3 FCDD provides full-resolution anomaly heatmaps. However, this upsampling involves the choice of $\sigma$ for the Gaussian kernel. In this section, we demonstrate the effect of this hyperparameter on the explanation performance of FCDD on MVTec-AD. Table 5 shows the pixel-wise mean AUC, Figure 11 corresponding heatmaps.

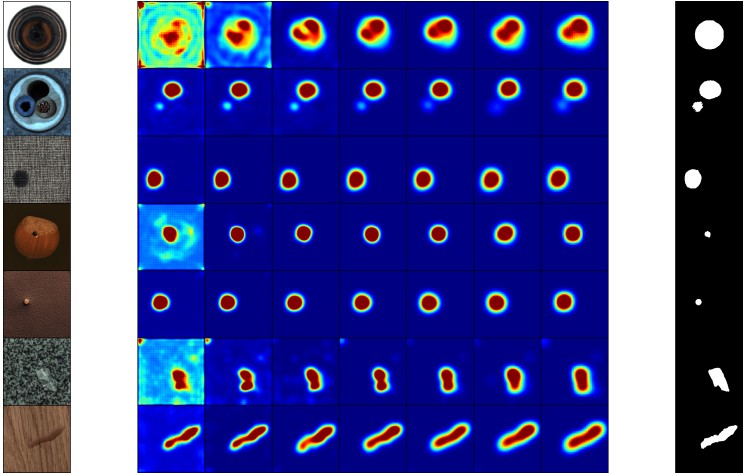

Figure 11: Anomaly heatmaps for seven anomalous test samples of MVTec-AD. We grow $\sigma$ from 4 (left) to 16 (right).

Table 5: Pixel-wise mean AUC (over all classes and 5 seeds per class) for MVTec-AD and different $\sigma$.

| $\sigma$ | 4 | 6 | 8 | 10 | 12 | 14 | 16 |
|---|---|---|---|---|---|---|---|
| AUC | 0.8567 | 0.8836 | 0.9030 | 0.9124 | 0.9164 | 0.9217 | 0.9208 |

## C    ANOMALY HEATMAP VISUALIZATION

For anomaly heatmap visualization, the FCDD anomaly scores $A'(X)$ need to be rescaled to values in $[0, 1]$. Instead of applying standard min-max scaling that would divide all heatmap entries by $\max A'(X)$, we use anomaly score quantiles to adjust the contrast in the heatmaps. For a collection of inputs $\mathcal{X} = \{X_1, \ldots, X_n\}$ with corresponding full-resolution anomaly heatmaps

$\mathcal{A} = \{A'(X_1), \ldots, A'(X_n)\}$, the normalized heatmap $I(X)$ for some $A'(X)$ is computed as

$$I(X)_j = \min\left\{ \frac{A'(X)_j - \min(\mathcal{A})}{q_\eta(\{A' - \min(\mathcal{A}) \mid A' \in \mathcal{A}\})}, 1 \right\},$$

where $j$ denotes the $j$-th pixel and $q_\eta$ the $\eta$-th percentile over all pixels and examples in $\mathcal{A}$. The subtraction and $\min$ operation are applied on a pixel level, i.e. the minimum is extracted over all pixels and all samples of $\mathcal{A}$ and subtraction is then applied elementwise. Using the $\eta$-th percentile might leave some of the values above 1, which is why we finally clamp the pixels at 1.

The specific choice of $\eta$ and set of samples $\mathcal{X}$ differs per figure. We select them to highlight different properties of the heatmaps. In general, the lower $\eta$ the more red (anomalous) regions we have in the heatmaps because more values are left above one (before clamping to 1) and vice versa. The choice of $\mathcal{X}$ ranges from just one sample $X$, such that $A'(X)$ is normalized only w.r.t. to its own scores (highlighting the most anomalous regions within the image), to the complete dataset (highlighting which regions look anomalous compared to the whole dataset). For the latter visualization we rebalance the dataset so that $\mathcal{X}$ contains an equal amount of nominal and anomalous images to maintain consistent scaling. The choice of $\eta$ and $\mathcal{X}$ is consistent per figure. In the following we list the choices made for the individual figures.

**MVTec-AD**    Figures 1, 10, and 11 use $\eta = 0.97$ and set $\mathcal{X}$ to $X$ for each heatmap $I(X)$ to show relative anomalies. So each image is normalized with respect to itself only.

**Fashion-MNIST**    Figure 4 uses $\eta = 0.85$ and sets $\mathcal{X}$ to the complete balanced test set.

**CIFAR-10**    Figures 6 and 9 use $\eta = 0.85$ and set $\mathcal{X}$ to $X$ for each heatmap $I(X)$ to show relative anomalies. So each image is normalized with respect to itself only.

**ImageNet**    Figure 5 uses $\eta = 0.97$ and sets $\mathcal{X}$ to the complete balanced test set.

**Pascal VOC**    Figure 8 uses $\eta = 0.99$ and sets $\mathcal{X}$ to the complete balanced test set.

**Heatmap Upsampling**    For the Gaussian kernel heatmap upsampling described in Algorithm 1, we set $\sigma$ to 1.2 for CIFAR-10 and Fashion-MNIST, to 8 for ImageNet and Pascal VOC, and to 12 for MVTec-AD.

## D    DETAILS ON THE NETWORK ARCHITECTURES

Here we provide the complete FCDD network architectures we used on the different datasets.

**Fashion-MNIST**

```
----------------------------------------------------------------
        Layer (type)              Output Shape          Param #
================================================================
           Conv2d-1          [-1, 128, 28, 28]           3,328
      BatchNorm2d-2          [-1, 128, 28, 28]             256
        LeakyReLU-3          [-1, 128, 28, 28]               0
        MaxPool2d-4          [-1, 128, 14, 14]               0
           Conv2d-5          [-1, 128, 14, 14]         409,728
        MaxPool2d-6            [-1, 128, 7, 7]               0
           Conv2d-7              [-1, 1, 7, 7]             129
================================================================
Total params: 413,441
Trainable params: 413,441
Non-trainable params: 0
Receptive field (pixels): 16 x 16
----------------------------------------------------------------
```

**CIFAR-10**

```
----------------------------------------------------------------
        Layer (type)              Output Shape         Param #
================================================================
            Conv2d-1          [-1, 128, 32, 32]          3,584
       BatchNorm2d-2          [-1, 128, 32, 32]            256
         LeakyReLU-3          [-1, 128, 32, 32]              0
         MaxPool2d-4          [-1, 128, 16, 16]              0
            Conv2d-5          [-1, 256, 16, 16]        295,168
       BatchNorm2d-6          [-1, 256, 16, 16]            512
         LeakyReLU-7          [-1, 256, 16, 16]              0
            Conv2d-8          [-1, 256, 16, 16]        590,080
       BatchNorm2d-9          [-1, 256, 16, 16]            512
        LeakyReLU-10          [-1, 256, 16, 16]              0
        MaxPool2d-11            [-1, 256, 8, 8]              0
           Conv2d-12            [-1, 128, 8, 8]        295,040
           Conv2d-13              [-1, 1, 8, 8]            129
================================================================
Total params: 1,185,281
Trainable params: 1,185,281
Non-trainable params: 0
Receptive field (pixels): 22 x 22
----------------------------------------------------------------
```

**ImageNet, MVTec-AD, and Pascal VOC**

```
----------------------------------------------------------------
        Layer (type)              Output Shape         Param #
================================================================
            Conv2d-1         [-1, 64, 224, 224]          1,792
       BatchNorm2d-2         [-1, 64, 224, 224]            128
             ReLU-3         [-1, 64, 224, 224]              0
         MaxPool2d-4         [-1, 64, 112, 112]              0
            Conv2d-5        [-1, 128, 112, 112]         73,856
       BatchNorm2d-6        [-1, 128, 112, 112]            256
             ReLU-7        [-1, 128, 112, 112]              0
         MaxPool2d-8          [-1, 128, 56, 56]              0
            Conv2d-9          [-1, 256, 56, 56]        295,168
      BatchNorm2d-10          [-1, 256, 56, 56]            512
            ReLU-11          [-1, 256, 56, 56]              0
           Conv2d-12          [-1, 256, 56, 56]        590,080
      BatchNorm2d-13          [-1, 256, 56, 56]            512
            ReLU-14          [-1, 256, 56, 56]              0
        MaxPool2d-15          [-1, 256, 28, 28]              0
           Conv2d-16          [-1, 512, 28, 28]      1,180,160
      BatchNorm2d-17          [-1, 512, 28, 28]          1,024
            ReLU-18          [-1, 512, 28, 28]              0
           Conv2d-19          [-1, 512, 28, 28]      2,359,808
      BatchNorm2d-20          [-1, 512, 28, 28]          1,024
            ReLU-21          [-1, 512, 28, 28]              0
           Conv2d-22            [-1, 1, 28, 28]            513
================================================================
Total params: 4,504,833
Trainable params: 4,504,833
Non-trainable params: 0
Receptive field (pixels): 62 x 62
----------------------------------------------------------------
```

# E    TRAINING AND OPTIMIZATION

Here we provide the training and optimization details for the individual experiments from Section 4.

We apply common pre-processing (e.g. data normalization) and data augmentation steps in our data loading pipeline. To sample auxiliary anomalies in an online manner during training, each nominal sample of a batch has a 50% chance of being replaced by a randomly picked auxiliary anomaly. This leads to balanced training batches for sufficiently large batch sizes. One epoch in our implementation still refers to the original nominal data training set size, so that approximately 50% of the nominal samples have been seen per training epoch. Below, we list further details for the specific datasets.

**Fashion-MNIST**   We train for 400 epochs using a batch size of 128 samples. We optimize the network parameters using SGD (Bottou, 2010) with Nesterov momentum ($\mu = 0.9$) (Sutskever et al., 2013), weight decay of $10^{-6}$ and an initial learning rate of 0.01, which decreases the previous

learning rate per epoch by a factor of $0.98$. The pre-processing pipeline is: (1) Random crop to size 28 with beforehand zero-padding of 2 pixels on all sides (2) random horizontal flipping with a chance of 50% (3) data normalization.

**CIFAR-10**    We train for 600 epochs using a batch size of 200 samples. We optimize the network using Adam (Kingma and Ba, 2015) ($\beta = (0.9, 0.999)$) with weight decay $10^{-6}$ and an initial learning rate of 0.001 which is decreased by a factor of 10 at epoch 400 and 500. The pre-processing pipeline is: (1) Random color jitter with all parameters[3] set to 0.01 (2) random crop to size 32 with beforehand zero-padding of 4 pixels on all sides (3) random horizontal flipping with a chance of 50% (4) additive Gaussian noise with $\sigma = 0.001$ (5) data normalization.

**ImageNet**    We use the same setup as in CIFAR-10, but resize all images to size $256{\times}256$ before forwarding them through the pipeline and change the random crop to size 224 with no padding. Test samples are center cropped to a size of 224 before being normalized.

**Pascal VOC**    We use the same setup as in CIFAR-10, but resize all images to size $224{\times}224$ before forwarding them through the pipeline and remove the Random Crop step.

**MVTec-AD**    For MVTec-AD we redefine an epoch to be ten times an iteration of the full dataset because this improves the computational performance of the data pipeline. We train for 200 epochs using SGD with Nesterov momentum ($\mu = 0.9$), weight decay $10^{-4}$, and an initial learning rate of 0.001, which decreases per epoch by a factor of 0.985. The pre-processing pipeline is: (1) Resize to $240{\times}240$ pixels (2) random crop to size 224 with no padding (3) random color jitter with either all parameters set to 0.04 or 0.0005, randomly chosen (4) 50% chance to apply additive Gaussian noise (5) data normalization.

# F    QUANTITATIVE DETECTION RESULTS FOR INDIVIDUAL CLASSES

Table 6 shows the class-wise results on Fashion-MNIST for AE, Deep Support Vector Data Description (DSVDD) (Ruff et al., 2018; Bergman and Hoshen, 2020) and Geometric Transformation based AD (GEO) (Golan and El-Yaniv, 2018).

Table 6: AUC scores for all classes of Fashion-MNIST (Xiao et al., 2017).

|  | | without OE | | with OE |
|---|---|---|---|---|
|  | AE | DSVDD* | Geo* | FCDD |
| T-Shirt/Top | 0.85 | 0.98 | 0.99 | 0.82 |
| Trouser | 0.91 | 0.90 | 0.98 | 0.98 |
| Pullover | 0.78 | 0.91 | 0.91 | 0.84 |
| Dress | 0.88 | 0.94 | 0.90 | 0.92 |
| Coat | 0.88 | 0.89 | 0.92 | 0.87 |
| Sandal | 0.45 | 0.92 | 0.93 | 0.90 |
| Shirt | 0.70 | 0.83 | 0.83 | 0.75 |
| Sneaker | 0.96 | 0.99 | 0.99 | 0.99 |
| Bag | 0.87 | 0.92 | 0.91 | 0.86 |
| Ankle Boot | 0.96 | 0.99 | 0.99 | 0.94 |
| Mean | 0.82 | 0.93 | 0.94 | 0.89 |

In Table 7 the class-wise results for CIFAR-10 are reported. Competitors without OE are AE (Ruff et al., 2018), DSVDD (Ruff et al., 2018), GEO (Golan and El-Yaniv, 2018) and an adaptation of GEO (GEO+) (Hendrycks et al., 2019b). Competitors with OE are the focal loss classifier (Hendrycks et al., 2019b), again GEO+ (Hendrycks et al., 2019b), Deep Semi-supervised Anomaly Detection (Deep SAD) (Ruff et al., 2020b;a) and the hypersphere Classifier (Ruff et al., 2020a).

In Table 8 the class-wise results for Imagenet are shown, where competitors are the AE, the focal loss classifier (Hendrycks et al., 2019b), Geo+ (Hendrycks et al., 2019b), Deep SAD (Ruff et al., 2020b) and HSC (Ruff et al., 2020a). Results from the literature are marked with an asterisk.

---

[3]https://pytorch.org/docs/1.4.0/torchvision/transforms.html#torchvision.transforms.ColorJitter

Table 7: AUC scores for all classes of CIFAR-10 (Krizhevsky et al., 2009).

| | without OE | | | | with OE | | | | |
| | AE* | DSVDD* | GEO* | Geo+* | Focal* | Geo+* | Deep SAD* | HSC* | FCDD |
|---|---|---|---|---|---|---|---|---|---|
| Airplane | 0.59 | 0.62 | 0.75 | 0.78 | 0.88 | 0.90 | 0.94 | 0.97 | 0.95 |
| Automobile | 0.57 | 0.66 | 0.96 | 0.97 | 0.94 | 0.99 | 0.98 | 0.99 | 0.96 |
| Bird | 0.49 | 0.51 | 0.78 | 0.87 | 0.79 | 0.94 | 0.90 | 0.93 | 0.91 |
| Cat | 0.58 | 0.59 | 0.72 | 0.81 | 0.80 | 0.88 | 0.87 | 0.90 | 0.90 |
| Deer | 0.54 | 0.61 | 0.88 | 0.93 | 0.82 | 0.97 | 0.95 | 0.97 | 0.94 |
| Dog | 0.62 | 0.66 | 0.88 | 0.90 | 0.86 | 0.94 | 0.93 | 0.94 | 0.93 |
| Frog | 0.51 | 0.68 | 0.83 | 0.91 | 0.93 | 0.97 | 0.97 | 0.98 | 0.97 |
| Horse | 0.59 | 0.67 | 0.96 | 0.97 | 0.88 | 0.99 | 0.97 | 0.98 | 0.96 |
| Ship | 0.77 | 0.76 | 0.93 | 0.95 | 0.93 | 0.99 | 0.97 | 0.98 | 0.97 |
| Truck | 0.67 | 0.73 | 0.91 | 0.93 | 0.92 | 0.99 | 0.96 | 0.97 | 0.96 |
| Mean | 0.59 | 0.65 | 0.86 | 0.90 | 0.87 | 0.96 | 0.95 | 0.96 | 0.95 |

Table 8: AUC scores for 30 classes of ImageNet (Deng et al., 2009).

| | without OE | | with OE | | | |
| | AE | Focal* | Geo+* | Deep SAD* | HSC* | FCDD |
|---|---|---|---|---|---|---|
| Acorn | 0.45 | × | × | 0.99 | 0.99 | 0.97 |
| Airliner | 0.80 | × | × | 0.97 | 1.00 | 0.98 |
| Ambulance | 0.25 | × | × | 0.99 | 1.00 | 0.99 |
| American alligator | 0.61 | × | × | 0.93 | 0.98 | 0.97 |
| Banjo | 0.45 | × | × | 0.97 | 0.98 | 0.91 |
| Barn | 0.59 | × | × | 0.99 | 1.00 | 0.97 |
| Bikini | 0.46 | × | × | 0.97 | 0.99 | 0.94 |
| Digital clock | 0.63 | × | × | 0.99 | 0.97 | 0.92 |
| Dragonfly | 0.62 | × | × | 0.99 | 0.98 | 0.98 |
| Dumbbell | 0.42 | × | × | 0.93 | 0.92 | 0.88 |
| Forklift | 0.28 | × | × | 0.91 | 0.99 | 0.94 |
| Goblet | 0.63 | × | × | 0.92 | 0.94 | 0.90 |
| Grand piano | 0.45 | × | × | 1.00 | 0.97 | 0.95 |
| Hotdog | 0.48 | × | × | 0.96 | 0.99 | 0.97 |
| Hourglass | 0.58 | × | × | 0.96 | 0.97 | 0.92 |
| Manhole cover | 0.70 | × | × | 0.99 | 1.00 | 1.00 |
| Mosque | 0.72 | × | × | 0.99 | 0.99 | 0.97 |
| Nail | 0.57 | × | × | 0.93 | 0.94 | 0.92 |
| Parking meter | 0.45 | × | × | 0.99 | 0.93 | 0.87 |
| Pillow | 0.40 | × | × | 0.99 | 0.94 | 0.94 |
| Revolver | 0.60 | × | × | 0.98 | 0.98 | 0.93 |
| Rotary dial telephone | 0.58 | × | × | 0.90 | 0.98 | 0.91 |
| Schooner | 0.65 | × | × | 0.99 | 0.99 | 0.96 |
| Snowmobile | 0.54 | × | × | 0.98 | 0.99 | 0.97 |
| Soccer ball | 0.46 | × | × | 0.97 | 0.93 | 0.86 |
| Stingray | 0.84 | × | × | 0.99 | 0.99 | 0.97 |
| Strawberry | 0.44 | × | × | 0.98 | 0.99 | 0.97 |
| Tank | 0.57 | × | × | 0.97 | 0.99 | 0.96 |
| Toaster | 0.59 | × | × | 0.98 | 0.92 | 0.79 |
| Volcano | 0.90 | × | × | 0.90 | 1.00 | 0.97 |
| Mean | 0.56 | 0.56 | 0.86 | 0.97 | 0.97 | 0.94 |

## G    FURTHER QUALITATIVE ANOMALY HEATMAP RESULTS

In this section we report some further anomaly heatmaps, unblurred baseline heatmaps, as well as class-wise heatmaps for all datasets.

**Unblurred Anomaly Heatmap Baselines**    Here we show unblurred baseline heatmaps for the figures in Section 4.1. Figures 12, 13, and 14 show the unblurred heatmaps for Fashion-MNIST, ImageNet, and CIFAR-10 respectively.

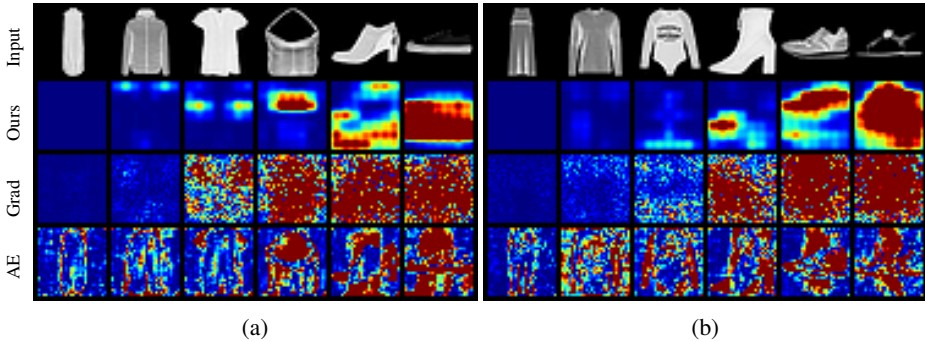

Figure 12: Anomaly heatmaps for anomalous test samples of a Fashion-MNIST model trained on nominal class "trousers." In (a) CIFAR-100 was used for OE and in (b) EMNIST.

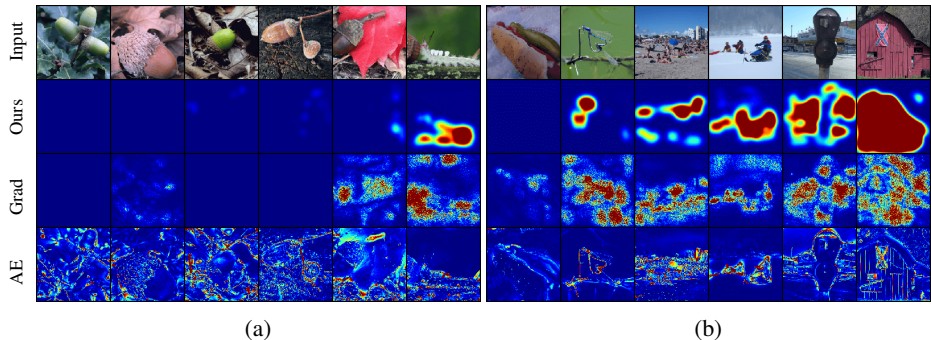

Figure 13: Anomaly heatmaps of an ImageNet model trained on nominal class "acorns." (a) are nominal and (b) anomalous samples.

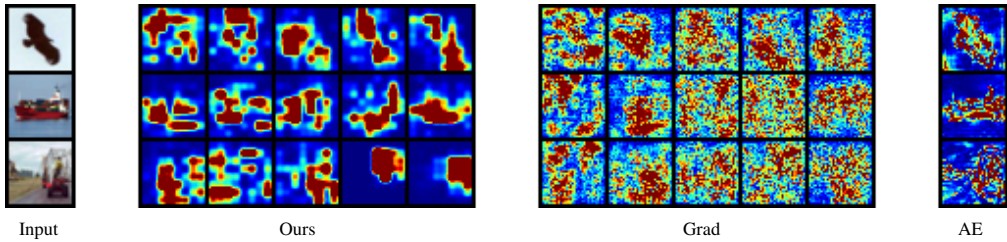

Figure 14: Anomaly heatmaps for three anomalous test samples (Input left) on a CIFAR-10 model trained on nominal class "airplane." The second, third, and fourth blocks show the heatmaps of FCDD (Ours), gradient-based heatmaps of HSC, and AE heatmaps respectively. For Ours and Grad, we grow the number of OE samples from 2, 8, 128, 2048 to full OE. AE is not able to incorporate OE.

**Class-wise Anomaly Heatmaps**    Due to space restrictions we have only shown heatmaps for some of the classes in the main paper. Here we also report a collection of heatmaps for all classes.

We show heatmaps with adjusted contrast curves by setting $\mathcal{X}$ to the balanced set of all samples for all datasets in this section. Further, we set $\eta = 0.85$ for Fashion-MNIST and CIFAR-10, $\eta = 0.99$ for MVTec-AD, and $\eta = 0.97$ for ImageNet. Note that, to keep the heatmaps for different classes comparable, we use a unified normalization for all heatmaps in one figure. However, since for each class a separate anomaly detector is trained, this yields suboptimal visualizations for some of the classes (for example, the "toothbrush" images for MVTec-AD in Figure 18 where the heatmaps just show a huge red blob). Tweaking the normalization for such classes reveals that the heatmaps actually tend to mark the correct anomalous regions, which in the case of "toothbrushes" can be seen in the explanation performance evaluation in Table 2.

The rows in all heatmaps show the following: (1) Input samples (2) FCDD heatmaps (3) gradient heatmaps with HSC (4) autoencoder reconstruction heatmaps. Heatmaps for MVTec-AD add a fifth row containing the ground-truth anomaly map.

Heatmaps for Fashion-MNIST using auxiliary anomalies from CIFAR-100 are in Figure 15, using EMNIST for OE instead are in Figure 16. CIFAR-10 heatmaps are in Figure 17, and heatmaps for all classes of MVTec-AD are in Figure 18. Finally, we present ImageNet heatmaps in Figures 19 and 20.

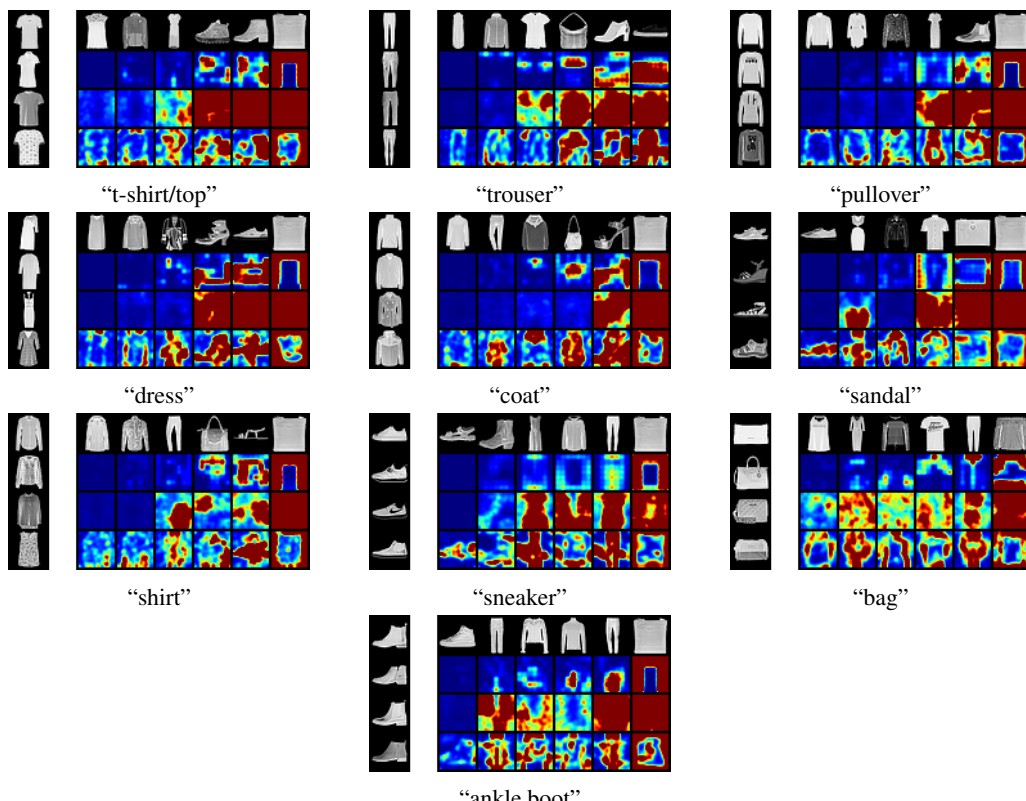

Figure 15: Anomaly heatmaps for anomalous test samples in Fashion-MNIST using CIFAR-100 OE. Columns are ordered by increasing anomaly score from left to right. The subcaptions refer to the nominal class that each model is trained on, for which some examples are also displayed as a separate column on the left.

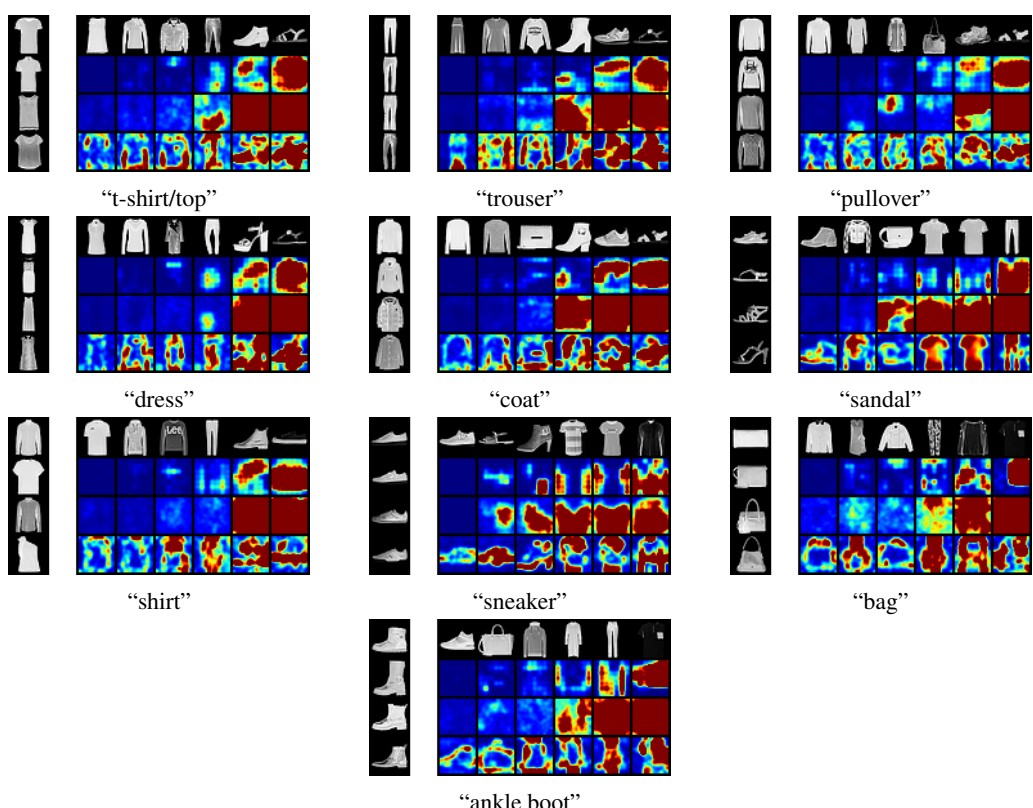

Figure 16: Anomaly heatmaps for anomalous test samples in Fashion-MNIST using EMNIST OE. Columns are ordered by increasing anomaly score from left to right. The subcaptions refer to the nominal class that each model is trained on, for which some examples are also displayed as a separate column on the left.

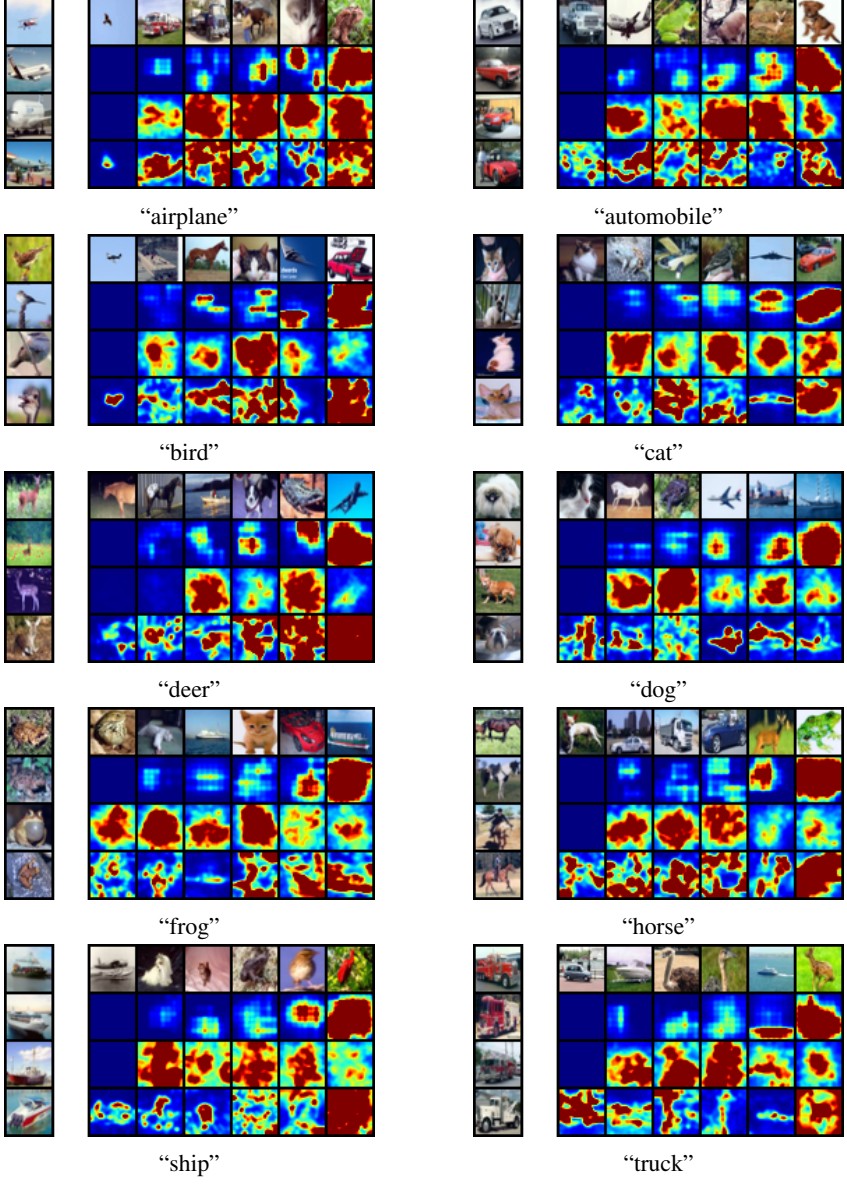

Figure 17: Anomaly heatmaps for anomalous test samples in CIFAR-10. Columns are ordered by increasing anomaly score from left to right. The subcaptions refer to the nominal class that each model is trained on, for which some examples are also displayed as a separate column on the left.

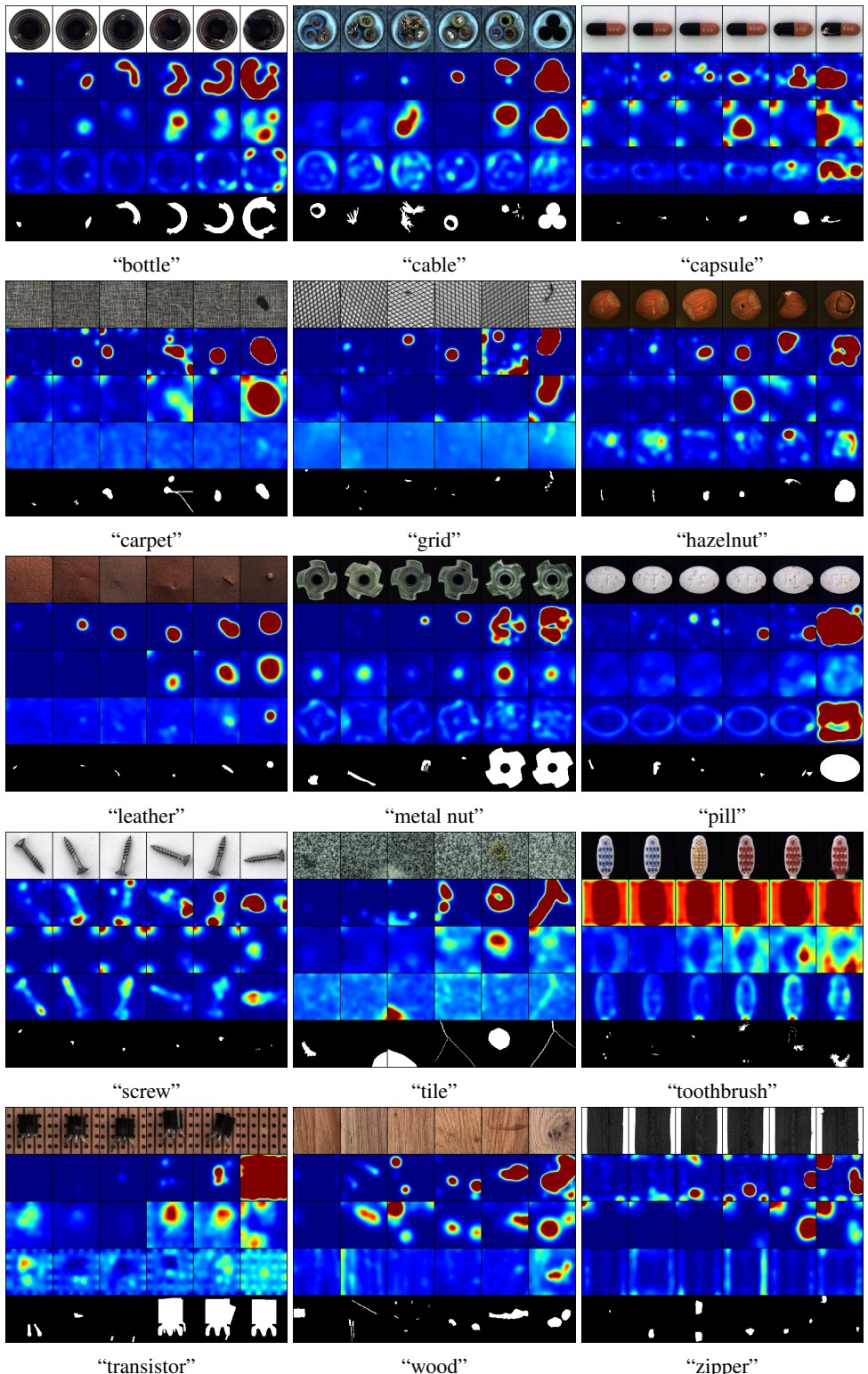

Figure 18: Anomaly heatmaps for anomalous test samples in MVTec-AD. Columns are ordered by increasing anomaly score from left to right. The subcaptions refer to the nominal class that each model is trained on.

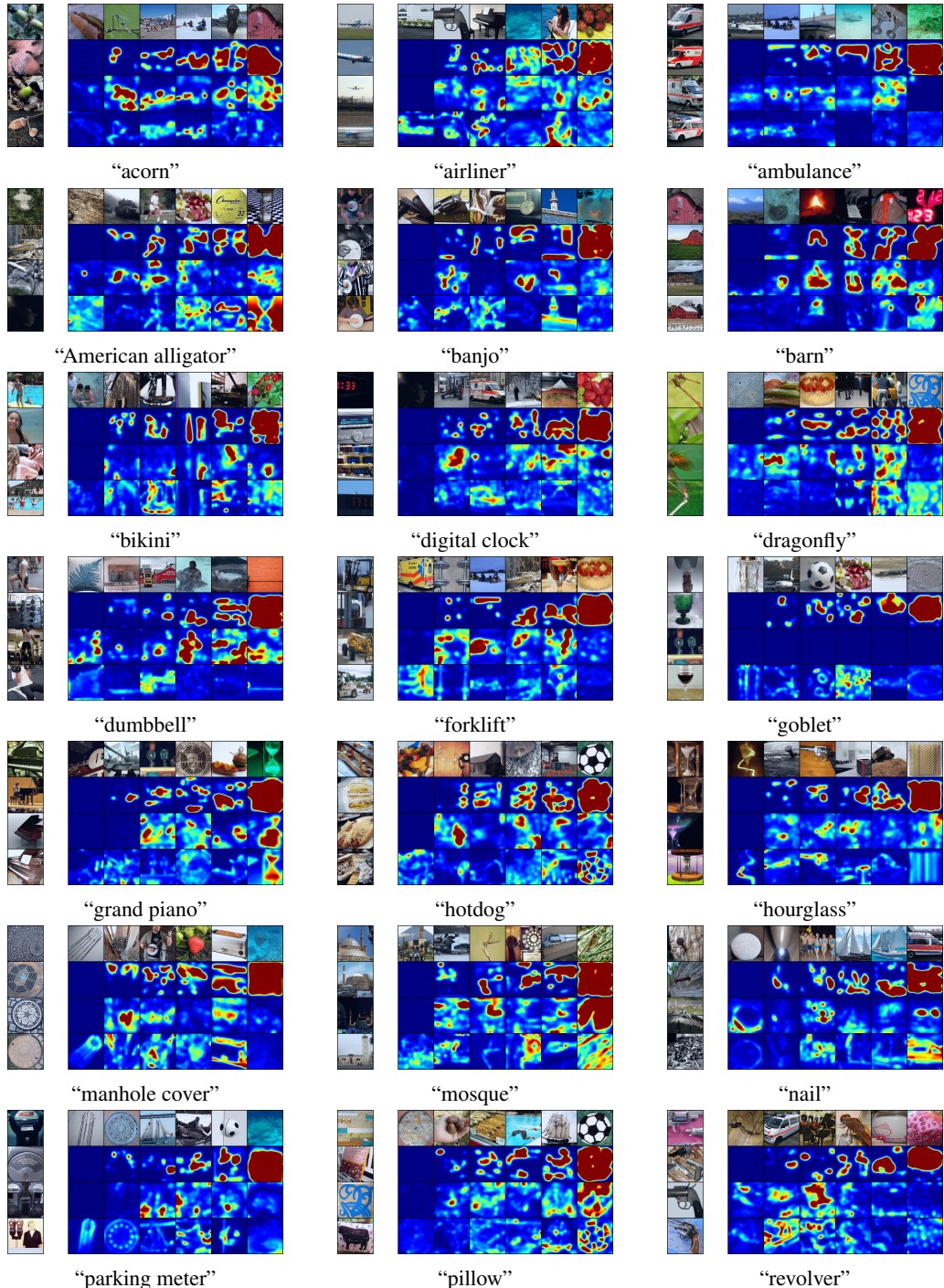

Figure 19: Anomaly heatmaps for anomalous test samples in ImageNet, where classes 1-21 are shown. Columns are ordered by increasing anomaly score from left to right. The subcaptions refer to the nominal class that each model is trained on, for which some examples are also displayed as a separate column on the left.

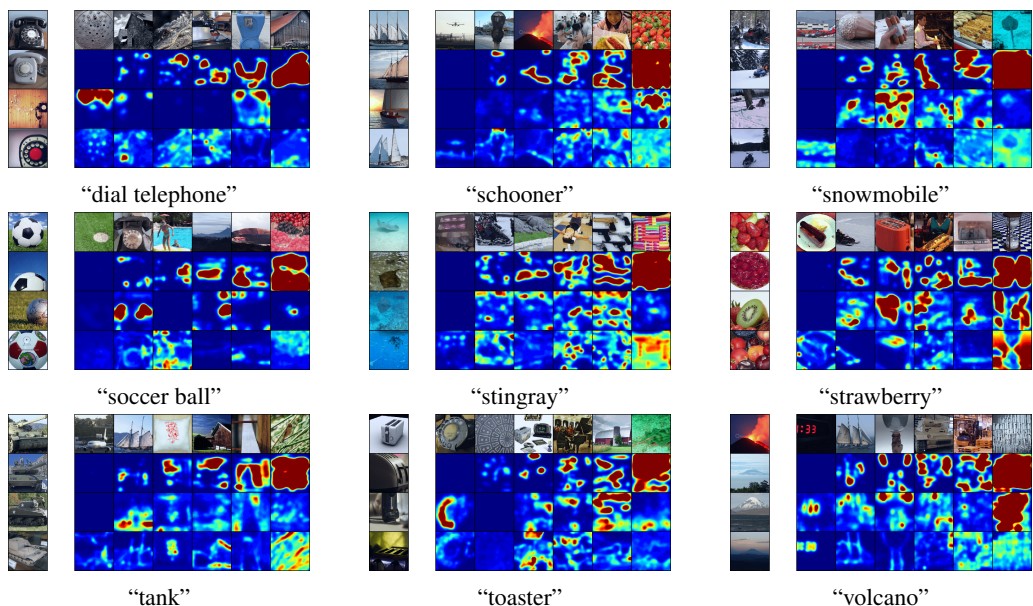

Figure 20: Anomaly heatmaps for anomalous test samples in ImageNet, where classes 22-30 are shown. Columns are ordered by increasing anomaly score from left to right. The subcaptions refer to the nominal class that each model is trained on, for which some examples are also displayed as a separate column on the left.

