# OpenReview forum: "Explainable Deep One-Class Classification"
_ICLR.cc/2021/Conference — ICLR 2021 Poster_

### Official Review · AnonReviewer2 · 2020-10-26
**An elegant and simple approach to deep one-class classification that provides explainable decisions**

**Rating:** 7
**Confidence:** 4

**Review:**

This paper presents a one-class classification method using a fully convolutional model and directly using the output map as an explanation map. The method is dubbed FCDD for fully convolutional data descriptor. FCDD uses a hypersphere classifier combined with a pseudo-Huber loss. FCDD is trained using outliers exposure (OE) from a different but related dataset. The empirical study consists of 3 parts:
 * a quantitative comparison of anomaly detection (AD) with SOT methods on three datasets: Fashion-MNIST, CIFAR-10 and ImageNET. The proposed method is shown to be competitive with SOT.
 * a qualitative study showing that the explanation provided by FCDD is more useful compared to other methods.
 * a quantitative comparison of the explanation performance on the MVTec-AD dataset. This shows that FCDD beats other SOT approaches and can also be easily adapted to accept actual anomalies examples, which results in even improved numbers.

PROS:

 * The paper is very well written (excellent grammar!) and easy to follow, with clear, complete and concise explanations.

 * The method is simple and elegant. It should be easy to reproduce it.

 * The results are competitive with SOT for AD and beats it for the quality explanation (with the caveat below).

CONS:

 * The novelty is incremental: Combining fully-convolutional CNN with a hypersphere classifier.

 * The comparison with other methods for MVTEC is only with published results. FCDD uses a confetti noise to create OE examples. However, it is not clear which OE methods (if any) were used in the published results. Therefore it is not clear if the improved SOT comes from a better OE method or from the FCDD itself.

 *  The paper also describes a fixed-kernel (Gaussian) upsampling method using strided transposed convolutions, which I find unnecessary to be added to the paper.

 * the qualitative examples on figure 1 should be compared with other methods.

 * fig 5 should show the nominal class for visual reference.


 Overall this is a well written paper, presenting a simple and reproducible approach to explainable AD that beats the SOT.

---

> ### Author Response · Authors · 2020-11-21
> **Author Response to Reviewer 2**
>
> Thank you for your review, we will incorporate your suggestions into the camera-ready version. We address a few points below.
>
> > **"fig 5 should show the nominal class for visual reference"** \
> Figure 5 shows heatmaps for a model trained on the "acorn" class where (a) shows anomalous samples (no acorns) and (b) nominal ones (acorns). Therefore, we already show the nominal class for "visual reference" in this figure. We will also add nominal examples for the other figures in the camera-ready version.
>
> > **"the qualitative examples on figure 1 should be compared with other methods"** \
> We provide heatmaps for all other methods on all MVTec-AD classes in Figure 18.
>
> > **"The comparison with other methods for MVTEC is only with published results. FCDD uses a confetti noise to create OE examples. However, it is not clear which OE methods (if any) were used in the published results. Therefore it is not clear if the improved SOT comes from a better OE method or from the FCDD itself."** \
> Unlike standard OE we do not use an auxiliary dataset, which we agree would be an unfair advantage. We instead introduce a noise model which we tried to make general-purpose. We are unaware of other explainable anomaly detection methods with OE on MVTec-AD and we seem to be the first to incorporate an OE-style approach to explainable anomaly detection.

---

### Official Review · AnonReviewer1 · 2020-10-28
**This paper addresses the topic of explainability in Deep One-Class Classification. A new method, Fully Convolutional Data Description (FCDD), is presented in which the mapped samples themselves are explainable heatmaps. It is shown that FCDD can provide transparent explanations of anomalies while being better or close to the state of the art on standard AD-benchmarks.**

**Rating:** 8
**Confidence:** 4

**Review:**

### **Evaluation**
- The paper is tackling the topic of explainable Deep One-Class-Classification. In contrast to comparable methods, like for example gradient-based heatmaps, FCDD has a spatial context. This fact facilitates the interpretability.
- The approach is well-motivated and compares well to the state of the art AD-methods.
- The paper provides sophisticated theoretical as well as empirical insights.


### **Detailed Comments for Authors**
- The visualisations given in the paper are well suited for the addressed problem. The Appendix gives a good insight into the functioning of the FCDD as well as the decision process of the networks. However, in figure 18. the output for "toothbrush" is shown a huge anomaly blob for all samples, while the ground truth shows just minimal areas. Why is that the case?
In contrast to this, the mean AUC in table 2 is 0.94 (and 0.95 for semi-supervised FCDD), which is quite good comparing to the visual impression.
- The section addressing the "Clever Hans" effect was a good addition, since it shows another perspective of the usage of the methods while clearly showing the learned characteristics. It might be beneficial to explain in more detail, why using the "horse" class as anomaly class, since it is kind of in inverse logic (it makes sense, but while reading one had to think twice about it).
- For the heatmap-overviews given, it might be valuable to have a comparison of one sample of the nominal class and multiple of the anomaly one. In this way, the trained class is shown and directly comparable to the anomalies.

#### **Strong points of the submission.**
- Very well written paper, which is easy to follow, based is based not least on good visualisation.
- The paper provides a good and detailed description of the theoretical concepts.
- It is positive that the examples also pointing some weaknesses of FCDD, like not recognising the anomaly, while simultaneously comparing to other methods. The method shows its strength in the spacial context. Given the extended appendix, the authors deliver a critical view on the possibilities as well as the potential limitations.

#### **Weak points of the submission.**
- the conclusion is quite sparse, and even missing an own section.

---

> ### Author Response · Authors · 2020-11-21
> **Author Response to Reviewer 1**
>
> Regarding the output for "toothbrushes": Since for each class a separate anomaly detector is trained, one would have to adjust the normalization for each class to fit the distribution of the anomaly scores (cf., Appendix C). However, to keep the heatmaps comparable, we chose to use a unified normalization for all heatmaps in one figure. Tweaking the normalization for the "toothbrush" images shows that the heatmaps actually tend to mark the correct anomalous regions. We will include an explanation for the camera-ready version.
>
> Thank you for your other suggestions on further improving clarity. We found them helpful and will include them in the camera-ready version.

---

### Official Review · AnonReviewer3 · 2020-10-29
**Recommendation to Reject**

**Rating:** 4
**Confidence:** 1

**Review:**

##########################################################################
Summary:

This paper presents a modified approach, on top of an existing method (DSVDD), for anomaly detection with a state of the art achievement in the unsupervised setting. In my opinion, the model itself is not a novel innovation. However, it fails when sprious image features exist.


##########################################################################
Reasons for score:

This paper has evidence to show better performance comparing to other, in terms of explanation.  But, however, to me personally, it doesn't meet a high standard, as the method is just a modification of what's already existing. And the heatmap as an explanation, demonstrated that model did pick up some superficial features for detection, such as color.  Lastly, as the authors also called out in the last session, it's not robust enough to overcome spurious features, which can be big problem for real applications.



##########################################################################
Questions during rebuttal period:

Please address and clarify the cons above

---

> ### Author Response · Authors · 2020-11-21
> **Author Response to Reviewer 3**
>
> Regarding "spurious" features: FCDD's core contribution is its ability to provide heatmaps as explanations in addition to the usual anomaly detection score, which makes FCDD's detection behavior transparent. For example, these explanations reveal the potential of one-class classification detectors to focus on image features which, while providing good discriminative power at training time, would not be desirable upon deployment/test time. A model has no way of knowing that such features are in fact undesirable and FCDD enables a practitioner to recognize and remedy (e.g. by cleaning or extending the training data) such behavior or other undesirable phenomena (e.g. to avoid unfair social bias).
>
> We will clarify this point in a camera-ready version and remark that other reviewers found the "Clever Hans" section to be a good addition.
> In light of this and the other reviews, we kindly ask you to reconsider your final score.

---

### Author Response · Authors · 2020-11-21
**General Author Response**

We thank all the reviewers for their helpful comments and are pleased that our work has been well received overall:

**R1: "The approach is well-motivated and compares well to the state of the art AD-methods."** \
**R1: "The paper provides sophisticated theoretical as well as empirical insights."** \
**R2: "The paper is very well written (excellent grammar!) and easy to follow, with clear, complete and concise explanations."** \
**R2: "The method is simple and elegant. It should be easy to reproduce it."**

There seem to be a few concerns from Reviewer 3 based on misunderstandings. We hope that our response and the other reviews help to alleviate these issues. We will also include and clarify these points in the final version. We will address the reviewers' comments individually.

---

### Decision · Program_Chairs · 2021-01-07
**Final Decision**

**Decision:**

Accept (Poster)

**Comment:**

The paper touches upon explainable anomaly detection. To that extend, it modified hypersphere classifier towards fully convolutional data description (FCDD). This is, as also pointed out by two of the reviewers a direct application of a fully convolutional network within the hyperspherical classifier. However, the paper  also shows how to then upsample the receptive field using a strided transposed convolution with a fixed Gaussian kernel. Both together with tackling explainable anomaly detection is important. Moreover, the empirical evaluation is quite exhaustive and shows several benefits compared to state-of-the-art. So, yes, incremental, but incremental for a very interesting an important case.